

# Sensitivity of cloud structure and precipitation to cloud microphysics schemes in ICON and implications for global km-scale simulations

Maor Sela[1], Philipp Weiss[2], and Philip Stier[1]

[1]Department of Physics, University of Oxford, Oxford, UK
[2]European Centre for Medium-Range Weather Forecasts, Bonn, Germany

**Correspondence:** Maor Sela (maor.sela@physics.ox.ac.uk)

**Abstract.** Cloud microphysics remains a major source of uncertainty in km-scale atmospheric models. While cloud-resolving models have advanced our understanding of cloud-climate interactions, their predictability remains limited. Most studies have examined either microphysics schemes or domain-size sensitivities, but their interactions are poorly understood. This study examines cloud structure and precipitation sensitivity to microphysics schemes and how they vary between regional and global

configurations within a single, consistent modelling framework. We analyse three convection-permitting simulations over the Amazon: two regional runs employing single- and double-moment microphysics schemes and a global single-moment run, with all other configurations consistent. We find that cloud hydrometeor characteristics are sensitive to the microphysics scheme. Specifically, the double-moment scheme produces up to five times more graupel and $100\%$ more rainfall, but twice as much cloud water and five times as much fog as the single-moment scheme. Despite these variations, precipitation, water vapour, and

outgoing longwave radiation remain consistent across schemes, suggesting large-scale constraints primarily govern integrated quantities. Furthermore, domain configuration amplifies sensitivities. The global simulation exhibits up to $150\%$ more fog and nearly double the cloud ice compared to the regional single-moment run, highlighting the role of large-scale circulation and lateral boundary conditions. These findings demonstrate that microphysics schemes influence cloud processes, while the domain setup determines how these sensitivities manifest. Improved observational constraints and perturbed-parameter ensembles are

therefore needed to evaluate model performance and separate tuning effects and structural uncertainty.

## 1   Introduction

Clouds are a key component of the atmospheric system, regulating the transport of heat, water, and momentum, while strongly influencing Earth's energy balance (Stephens, 2005). Their radiative and hydrological effects are controlled by micron-scale

physical processes, known as cloud microphysics, which govern the formation, growth, and sedimentation of hydrometeors. Due to their size, these interactions cannot be resolved and must be parameterised in atmospheric models. Uncertainties asso-



ciated with this parameterisation are among the leading contributors to biases in the representation of clouds (Morrison et al., 2020).

Cloud-resolving models (CRMs) have become central tools for improving process-level understanding of clouds and for refining parameterisations in coarser-scale models. Since CRMs operate at kilometre or sub-kilometre grid spacings, they can explicitly simulate convection and its dynamics. Hence, CRMs are widely used to study cloud–aerosol–radiation interactions and cloud–climate feedbacks, and are often referred to as numerical laboratories (e.g., Wyngaard, 2004; Guichard and Couvreux, 2017; Randall et al., 1996; Herbert et al., 2015; Khain et al., 2015; Hourdin et al., 2017; Dagan et al., 2018; Herbert et al., 2021). Despite their strengths, studies have shown that simulated clouds are sensitive to many factors, including cloud droplet number concentration, aerosol optical depth, seasonality, and specific model employed (Karydis et al., 2012; Costa-Surós et al., 2020; Christensen et al., 2023; Labbouz et al., 2018); furthermore, the magnitude of these responses strongly depends on model setup, such as, spatial resolution, domain size, and boundary conditions (Trenberth et al., 2009; Emanuel, 1994; Rao et al., 1988; Tao et al., 2012; Varble, 2018; Sherwood et al., 2020; Weigum et al., 2016; Li et al., 2024; Kipling et al., 2017; Marinescu et al., 2021; White et al., 2017; Zhang et al., 2017).

At the centre of these uncertainties lies the cloud microphysics parameterisation, which is implemented through microphysics schemes that approximate the statistical properties of hydrometeors (Adams-Selin et al., 2013; Morrison and Milbrandt, 2011; Tao et al., 2011). Different schemes, such as bin and bulk microphysics, employ different methods to evaluate hydrometeor statistics. Bulk microphysics schemes, the most common (Seiki et al., 2022), assume analytic functional forms for hydrometeor size distributions (e.g., gamma or lognormal) and predict a small set of statistical moments, such as mass mixing ratio or mean diameter (Lebo et al., 2012; Morrison and Milbrandt, 2011; Seifert and Beheng, 2006). This allows essential processes, including condensation, evaporation, deposition, and collision-coalescence, to be represented at reduced computational cost, while their complexity depends on the number of predicted moments.

Single-moment schemes predict only the mass mixing ratio of hydrometeor classes, typically cloud water, rain, ice, and snow. They are considered computationally efficient and reliable in km-scale models to reproduce the hydrological cycle and surface precipitation (Lebo et al., 2012). However, the prescribed number concentrations and fall speeds limit the representation of cloud variability, microphysical–radiative interactions, and phase partitioning (Khain et al., 2015). Double-moment schemes predict both mass and number concentration, allowing a more flexible treatment of processes such as growth, accretion, and riming (Seifert, 2011). This has been reported to improve the representation of hydrometeor spectra and precipitation development (e.g., Igel et al., 2015; Gettelman and Morrison, 2015). Beyond the added computational cost, double-moment schemes can introduce new biases from additional parameters and assumptions. For example, they can overestimate rainfall or condensate amounts in some environments (e.g., Wu and Petty, 2010; Van Weverberg et al., 2014).

As a result, differences between single- and double-moment schemes can significantly influence CRM outcomes, with several studies showing that microphysics affects simulated cloud fields as much as (or even more than) changes in resolution or external forcing (e.g., Igel et al., 2015; Guo et al., 2015; Khain et al., 2015; Seifert, 2011; Song and Zhang, 2011; Sullivan et al., 2016; Tao et al., 2016; White et al., 2017). Yet, with the current limitations of observational datasets, it remains difficult to determine which formulation performs more realistically.



Alongside the choice of microphysics, the choice of the domain also shapes CRM outcomes. Traditionally, CRMs are applied in limited-area domains, allowing for simulations with finer resolutions and longer integrations of specific weather events or localised phenomena, such as deep convection or regional precipitation patterns. Regional CRMs are particularly well-suited for process studies and case-specific experiments (Hohenegger et al., 2023), and have been widely used to investigate diurnal cycles, precipitation extremes, and mesoscale convective organisation. However, to constrain the atmospheric state and maintain consistency with observed large-scale conditions, regional simulations rely on lateral boundary conditions, typically derived from global reanalysis data or lower-resolution models. This can propagate biases or errors from driving data into the simulated domain, limiting its ability to represent the full evolution of large-scale circulations (Uchida et al., 2017; Radović et al., 2024). In addition, because regional simulations do not feed back to larger-scale flow, they decouple local processes from global circulation, thus preventing mutual interactions between local convection and large-scale dynamics (Guichard and Couvreux, 2017; Grabowski et al., 2000; Ibebuchi, 2022; Song et al., 2022).

Recent advances in high-performance computing have enabled CRMs to be employed globally, allowing km-scale simulations to explicitly resolve convective processes across the entire atmosphere. Global CRMs are increasingly used to explore large-scale circulation, climate sensitivity, and cloud–climate feedbacks (see, e.g., Herbert et al., 2024). In addition, global CRMs eliminate the need for lateral boundary conditions, resulting in a more consistent representation of large-scale phenomena, such as planetary waves and global circulation (Tomita et al., 2005; Hohenegger et al., 2023). However, the absence of external constraints can lead to divergence from observations, particularly for local weather systems (Hourdin et al., 2017); these setups may also struggle to capture small-scale processes due to resolution limitations. Global simulations also require a spin-up period, often around two weeks, to reach a steady state, during which transient imbalances can affect results (Tomita et al., 2005; Kodama et al., 2012; Stevens et al., 2019). As a consequence, these methodological differences between regional and global simulations can result in substantial divergence in their representation of clouds and precipitation (Rocheta et al., 2017; Ferrari et al., 2020).

In conclusion, the sensitivity of simulated clouds and precipitation to microphysics is well-documented, with differences between double-moment schemes exceeding those from aerosol perturbations (White et al., 2017), and higher resolutions have been shown to lead to improved mid-level cloud representation (Omanovic et al., 2024). By contrast, the role of domain configuration remains less explored, even though its limitations are increasingly recognised (Guichard and Couvreux, 2017; Grabowski et al., 2000; Song et al., 2022). Together, the uncertainties arising from microphysics schemes and from the choice between regional and global domains highlight the need to examine their interactions.

Motivated by these challenges, the present study investigates the sensitivity of cloud structure and precipitation to the microphysics scheme in a km-scale CRM and examines how these sensitivities extend from regional to global configurations. Specifically, we ask how simulated clouds and precipitation differ between single- and double-moment schemes, and what the implications are for representing these processes in global km-scale simulations. Our aim is not to provide a detailed analysis of individual microphysical rates or process interactions, but rather to identify the main discrepancies arising from different microphysics schemes and to examine how these uncertainties manifest across model configurations within the same framework.



The remainder of this paper is organised as follows. Section 2 describes the model setup and experimental design. Section 3 presents the results, beginning with a bulk comparison before focusing on specific sections of the vertical cloud structure. Section 4 discusses the interpretation of results, observational biases, and model limitations. Section 5 summarises the findings
and highlights their implications for future modelling efforts.

## 2   Methods

### 2.1   Model description

The ICOsahedral Non-hydrostatic modelling framework of the Max Planck Institute for Meteorology (ICON-MPIM), which simulates all Earth system components on an icosahedral-triangular C grid (Hohenegger et al., 2023) is used. ICON com-
prises three main components: ocean, land, and atmosphere. For the land surface, it employs the Jena Scheme for Biosphere-Atmosphere Coupling in Hamburg (JSBACH) version 4, providing boundary conditions such as albedo, roughness length, and flux parameters, while solving the surface energy balance coupled with atmospheric diffusion equations (Richtmyer and Morton, 1967). It features a multi-layer soil hydrology scheme for water storage (Hagemann and Stacke, 2015) and a hydrological discharge model for routing runoff into the oceans (Hagemann and Dümenil, 1997). In the atmospheric component, ICON-
MPIM utilises a hybrid sigma-z vertical coordinate system (the SLEVE scheme), incorporating a Rayleigh damping layer in the upper atmosphere (Leuenberger et al., 2010; Klemp et al., 2008). Within its regional configuration, a sponge layer is implemented along the lateral boundaries to prevent outward-propagating waves from reflecting, with the interior flow relaxed towards externally specified boundary data. In supersaturated regions, positive water vapour increments are cut to zero in the nudging zone to avoid an artificial increase in cloud water (DWD, 2020). ICON-MPIM employs radiation, cloud microphysics,
and turbulence schemes, deliberately avoiding parameterisation for shallow convection or subgrid-scale clouds to better resolve km-scale dynamics (Hohenegger et al., 2020). Its radiation scheme employs the Rapid Radiative Transfer Model for General circulation model applications-Parallel (RRTMGP) (Pincus et al., 2019), whereas turbulence is parameterised using a modified Smagorinsky scheme suited for cloud-resolving simulations (Dipankar et al., 2015). This configuration prioritises computational efficiency to explore Earth system dynamics and sensitivities to unresolved processes (Palmer and Stevens, 2019; Schär
et al., 2020). Cloud microphysical processes are represented using the single- and double-moment schemes of Baldauf et al. (2011) and Seifert and Beheng (2006), respectively.

The single-moment scheme predicts mass mixing ratios of water vapour, cloud water, rain, cloud ice, snow, and graupel. Warm-phase processes are represented by the parameterisation of Seifert and Beheng (2001), reduced to one-moment form by assuming a fixed cloud droplet number concentration of $N_c = 500\,\mathrm{cm}^{-3}$. Raindrop growth and sedimentation are modelled
using exponential size distributions with empirically derived terminal fall speeds. Cold-phase processes include graupel formation via raindrop freezing, cloud ice riming, and snow-to-graupel conversion when cloud water exceeds $0.2\,\mathrm{g\,kg}^{-1}$. Snow microphysics utilise temperature-dependent intercept parameters in exponential size distributions, enhancing the treatment of slower-falling aggregates at higher altitudes (Field et al., 2005). Graupel particles are assumed to have low density and moderate fall speeds based on the relationships of Heymsfield and Kajikawa (1987).




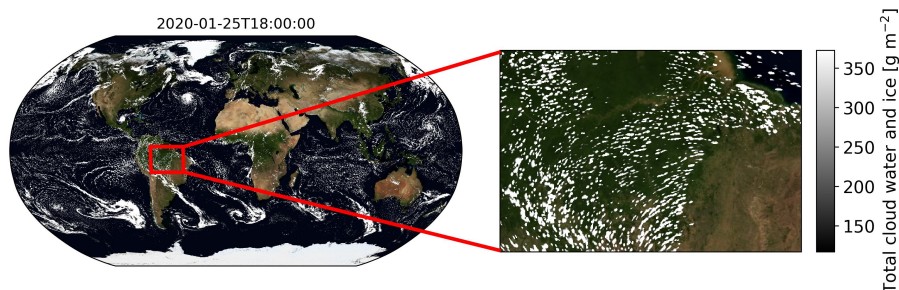

**Figure 1.** Liquid and ice water path simulated by ICON-MPIM. The left panel shows the global distribution, with a red box highlighting the Amazon domain analysed in this study. The background image is taken from NASA Earth Observatory (Stöckli et al., 2005).

The double-moment scheme predicts both mass and number concentrations of the same hydrometeors, including hail. Raindrop formation employs a stochastic bulk parameterisation, with autoconversion and accretion rates depending on both mass and number concentrations, making them sensitive to droplet size and concentration. Hydrometeor size spectra are represented with a modified gamma distribution, whose parameters are diagnosed from the prognosed mass and number concentrations. Cold-phase processes include size-dependent collection and freezing efficiencies. Graupel forms through riming of snow and

cloud ice, as well as melting and refreezing. Ice-phase interactions use collision efficiencies derived from Wisner et al. (1972), which account for the dependence of collection probability on particle size and relative fall speeds, providing a more realistic representation than the constant efficiencies assumed in simpler schemes. By prognosing number concentrations, the scheme explicitly represents processes such as ice multiplication, sedimentation velocities, and phase transitions under varying humidity and temperature conditions.

**2.2    Simulation setup**

To explore the sensitivity of cloud microphysics schemes, we compare two regional simulations using different microphysics schemes. To examine how these sensitivities interact with domain configuration, we also analyse a global simulation. All simulations are conducted using the R2B9 grid, which provides a horizontal grid spacing of approximately $5\,\mathrm{km}$. The atmosphere and land components use time steps of $10$ and $40\,\mathrm{s}$ in the regional and global setups, respectively, with radiation calculated

every 3 and $12\,\mathrm{min}$. The regional simulations use single- and double-moment microphysics schemes and are centred over the Amazon basin, spanning approximately $26°$ longitudinally ($68° - 42°\,$W) and $18°$ latitudinally ($15°\,$S - $3°\,$N). The global simulation employs the single-moment scheme and uses identical physical and numerical settings. From this simulation, the same Amazon region is selected (see Fig. 1) to enable direct comparison with the regional runs and to isolate the effects of domain configuration from those of microphysics representation. To ensure the global simulation reaches a statistically stable state

(Leduc and Laprise, 2009; Matte et al., 2017), a spin-up period of 12 days (from January 20 to 31, 2020) is included, with data collection beginning on the thirteenth day (1 February 2020). Hence, the experiment duration for which data was collected and analysed in all simulations was 1 to 7 February 2020.





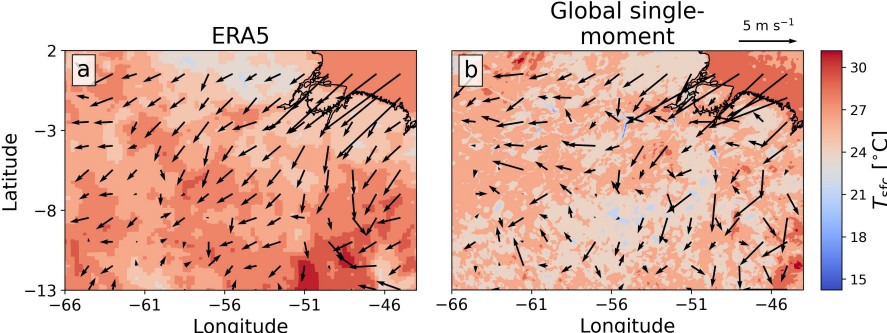

**Figure 2.** Surface temperature (shading) and wind field (arrows) on 1 February 2020, 00:00 UTC, from (a) the ERA5 reanalysis and (b) the global single-moment simulation.

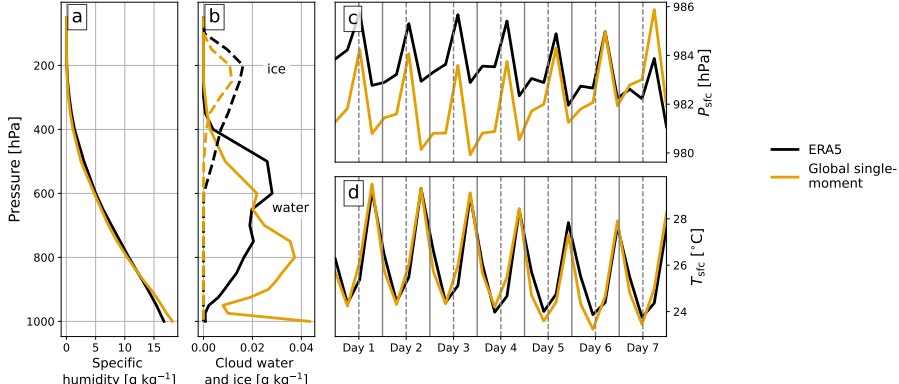

**Figure 3.** Comparison of ERA5 lateral boundary conditions (black curves), used to nudge the regional simulations, with corresponding data from the global single-moment simulation (yellow curves) sampled in the boundary zone. Panels (a) and (b) show the mean vertical profiles of (a) specific humidity and (b) cloud water (solid) and ice (dashed), and panels (c) and (d) show the time series of (c) surface pressure and (d) surface temperature.

ERA5 reanalysis data from the Copernicus Climate Change Service (Hersbach et al., 2017) is used to initialise all three simulations and provide lateral boundary conditions for the regional runs. Boundary conditions are updated every 6 hours and linearly interpolated between time intervals. Due to the spin-up period, the atmospheric states of the global and regional simulations diverge at the start of the experiment. As shown in Fig. 2, the global simulation (Fig. 2b) reproduces the main features of the ERA5 state (Fig. 2a), including surface wind direction over the ocean, sea surface temperature patterns, and the westward flow regime. Minor deviations are evident, characterised by slightly cooler surface temperatures and a weak divergence region in the near-surface winds.

To quantify differences at the domain boundary (the outermost 14 grid cells of the regional domain), Fig. 3 compares ERA5 boundary conditions with the same zone from the global simulation, matching the frequency of ERA5 input. The comparison





includes mean vertical profiles of specific humidity (Fig. 3a) and cloud water and ice (Fig. 3b), where solid lines indicate cloud water and dashed lines indicate cloud ice. Time series of surface pressure (Fig. 3c) and surface temperature (Fig. 3d) are also shown. Fig. 3 shows notable differences between ERA5 and the global single-moment simulation, including a near-surface specific humidity offset of about $2\,\mathrm{g\,kg^{-1}}$, pressure deviations up to $3\,\mathrm{hPa}$, and divergence in vertical cloud profiles. In the global simulation, there is substantially more low-level clouds. In ERA5, on the other hand, there are more mid-level and anvil clouds.

## 2.3 Observational data

To assess simulated precipitation rates and distributions, we use the Integrated Multi-satellitE Retrievals for Global Precipitation Measurement Mission (IMERG; 2020). IMERG estimates are derived from a combination of satellite observations and ground-based gauge measurements. Its high temporal ($30\,\mathrm{min}$) and spatial ($0.1°$) resolutions make it ideal for verification and comparison with CRM simulations. However, its evaluation for regional analysis is uncertain due to the lack of consistent ground-truth data (Li et al., 2023; Chen et al., 2020).

Nevertheless, IMERG effectively captures spatial precipitation patterns in regions with sparse rain gauges and complex topography (Hartke and Wright, 2022; Gilewski and Nawalany, 2018; Zhou et al., 2021), although its performance varies seasonally and geographically. Biases have been reported in the accuracy of winter precipitation over South America (Gadelha et al., 2019; da Silva et al., 2023), including misrepresentation of heavy rainfall and overestimation of moderate events (Talchabhadel et al., 2022; Muñoz de la Torre et al., 2024; Gadelha et al., 2019), and underdetection of low-intensity precipitation due to limitations of passive microwave retrieval (Bogerd et al., 2021). These issues are especially relevant in regions with strong seasonal variability, where the reliability of IMERG diminishes (Bulovic et al., 2020; Jiang and Bauer-Gottwein, 2019). Despite these limitations, IMERG remains valuable for evaluating simulated precipitation due to its high spatial and temporal resolution.

For outgoing longwave radiation (OLR), we use observations from the Advanced Baseline Imager (ABI) of the first satellite of the Geostationary Operational Environmental Satellites (GOES-16) series (Schmit and Gunshor, 2020). We estimate OLR from ABI level 2 in the cloud and moisture imagery product (MCMIP) channel $11.2\,\mu\mathrm{m}$ using the relationship from Ohring et al. (1984) between the observed longwave window brightness temperature (BT) and the flux equivalent BT.

## 3 Results

First, we examine the time-averaged distributions of cloud water and ice. Fig. 4 shows maps of liquid water path (LWP) and ice water path (IWP) across all three simulations. The distributions and magnitudes of LWP and IWP indicate sensitivity to the choice of the microphysics scheme. The single-moment simulation generally predicts higher LWP values, especially along the northeastern coast, while the double-moment simulation yields lower values. All simulations predict more clouds over and near the coastline; however, the global simulation shows clouds in the northern parts of the regions, whereas the regional simulations



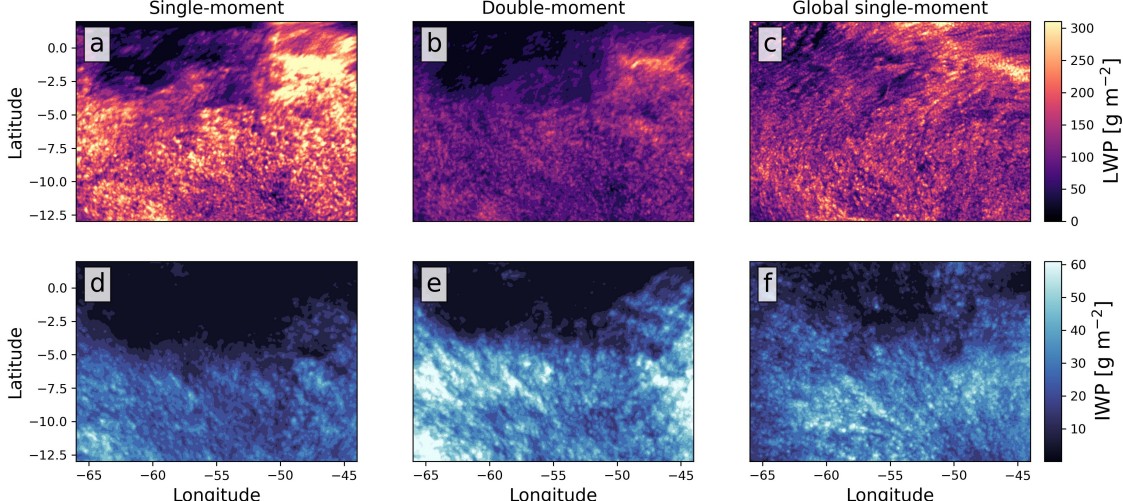

**Figure 4.** Time-averaged maps of the (a, d) liquid and ice water paths for the single-moment regional simulation, (b, e) for the double-moment regional simulation, and (c, f) for the global simulation.

remain relatively cloud-free. The double-moment simulation predicts a higher average IWP, with the global simulation showing more IWP in the northern "dry" regions as well (see Fig. 4d–f).

A closer examination of cloud structure reveals further differences. Fig. 5 presents the time- and domain-averaged mass mixing ratio profiles for cloud water, ice, rain, graupel, snow, and water vapour. For rain, graupel, and snow, the single-moment simulations (regional and global) predict similar profiles (Figs. 5c–e). In contrast, the double-moment simulation predicts up to twice the rain and six times the graupel compared to single-moment runs, while snow amounts are lower (Figs. 5c–e). Regarding cloud ice (Fig. 5b), the simulations show less divergence: the double-moment simulates a slightly wider vertical spread with peak values marginally higher than the regional single-moment but lower than the global one.

As for cloud ice, rain water, graupel, and snow, the cloud water profiles (Fig. 5a) indicate sensitivity to the microphysics scheme, particularly in low and mid-level clouds, with differences between the two single-moment simulations. For example, the global simulation predicts a lower cloud base, with higher vapour concentrations at lower levels. At around $750\,\mathrm{hPa}$, a distinction between low and mid-level clouds is evident in the global simulation. Moreover, the simulations disagree on the magnitude of cloud liquid water at the lowest model level above the surface (Fig. 5a). To further analyse this, we focus on this lowest layer, and, hereafter, define *fog* as the cloud water present within this layer.

## 3.1 Fog

Fig. 6 presents time-averaged maps of fog mass mixing ratios, reflecting the variability shown in Fig. 5a. The single-moment simulations (both regional and global) predict substantially more fog than the double-moment simulation, with the global simulation covering almost the entire domain and some clusters extending tens of kilometres horizontally. In contrast, the



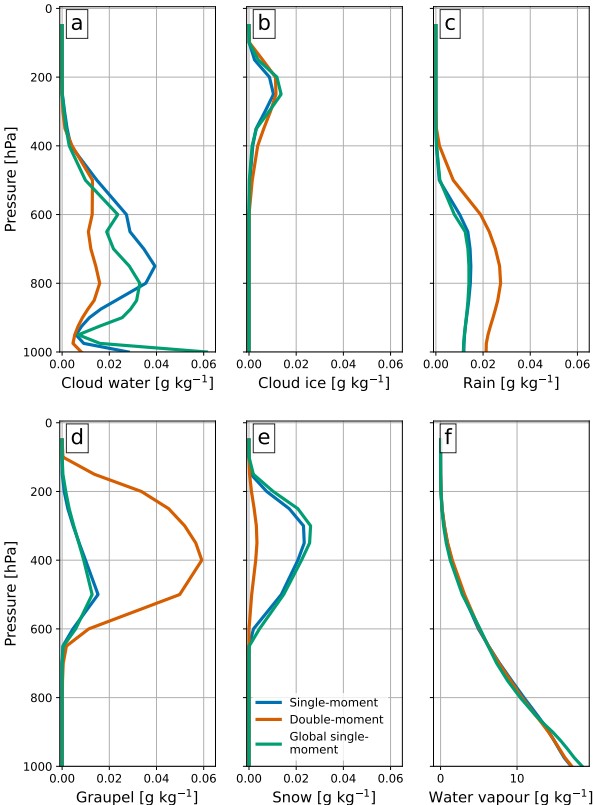

**Figure 5.** Vertical profiles mass mixing ratio of (a) cloud liquid water, (b) ice, (c) rain water, (d) graupel, (e) snow, and (f) water vapour for the regional single-moment (blue) and double-moment (red) simulations, shown alongside the global single-moment run (green). The profiles are averaged over the entire Amazon region and experiment duration.

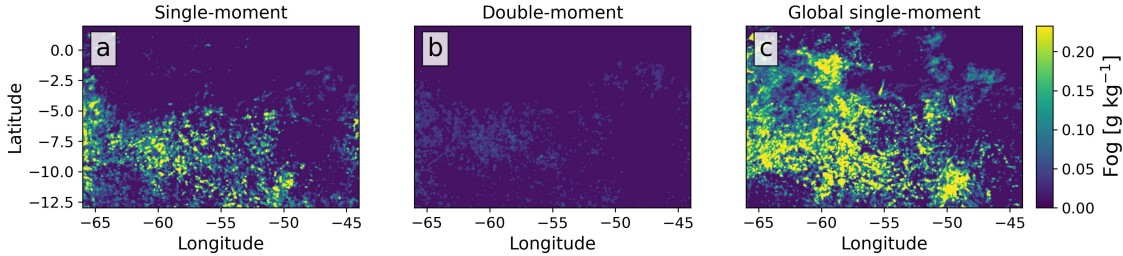

**Figure 6.** Time-averaged maps of fog mass mixing ratio for (a) the single-moment and (b) double-moment simulations, shown alongside (c) the global single-moment run.

double-moment simulation (Fig. 6b) shows the lowest amounts, as also indicated in Fig. 5a, suggesting that fog is more sensitive





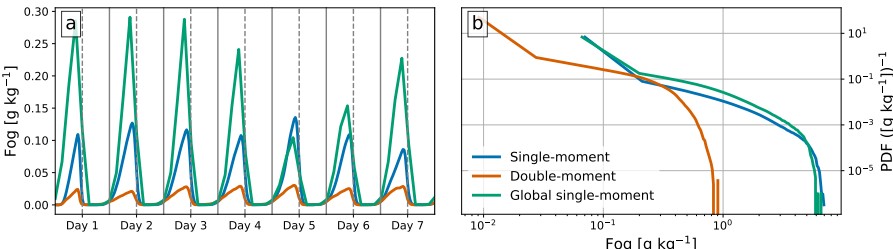

**Figure 7.** (a) Time series and (b) probability density functions of cloud liquid water mixing ratio at the lowest model level (fog) for the regional single-moment (blue) and double-moment (red) simulations, shown alongside the global single-moment run (green). The time series are averaged over the entire Amazon region and experiment duration.

to microphysics choices. In contrast, all simulations predict fog in the western region, which is strongly constrained by lateral boundary conditions.

As shown in Fig. 6, the domain-mean fog time series in Fig. 7a highlights significant differences in the fog diurnal cycle among the simulations. All three simulations exhibit similar diurnal cycles, starting around midnight UTC (9 pm local time) and lasting approximately 12 hours. However, amplitude differences are notable, with the double-moment simulation peaking at $0.025\,\mathrm{g\,kg^{-1}}$, the regional single-moment at $0.130\,\mathrm{g\,kg^{-1}}$, and the global single-moment at $0.295\,\mathrm{g\,kg^{-1}}$, with the global exhibiting greater variability. Interestingly, the global simulation shows a decrease in fog on day five, the only instance where its mean value falls below that of the regional simulation. The subsequent recovery by day seven coincides with changes in circulation patterns. Because the regional simulations are constrained by reanalysis data at their lateral boundaries, whereas the global run evolves freely, external influences outside the Amazon basin modify local conditions through internally generated dynamics. These changes enhance vertical mixing and horizontal transport, which remove moisture from the boundary layer and reduce fog formation. While the time series in Fig. 7a shows some similarities, the probability density function (PDF) in Fig. 7b (shown on log-log axes) highlights clearer differences. The single-moment simulations display comparable distributions, with more frequent fog occurrences across most of the range. However, the distribution predicted by the double-moment simulation is shifted to lower values compared to the regional simulations, with a maximum value reaching approximately $1\,\mathrm{g\,kg^{-1}}$. This highlights the significant influence of microphysics schemes on fog intensity over the Amazon.

## 3.2 Liquid clouds

Analysis of the warm-phase processes reveals time series differences in water vapour path, LWP, and rain water path across all three simulations (Fig. 8). While all simulations agree on the general cycle, differences in amplitude are evident. The regional simulations show nearly identical water vapour distributions (Fig. 8a), with differences less than $0.5\%$, whereas the global simulation exhibits levels up to $5\%$ higher. For LWP and rain (Figs. 8b and 8c), the single-moment simulations remain closely aligned throughout most of the experiment, while the double-moment simulation shows significantly lower LWP (up to twice less) and higher rain values (up to twice more). The global simulation also demonstrates greater diurnal variability in LWP,



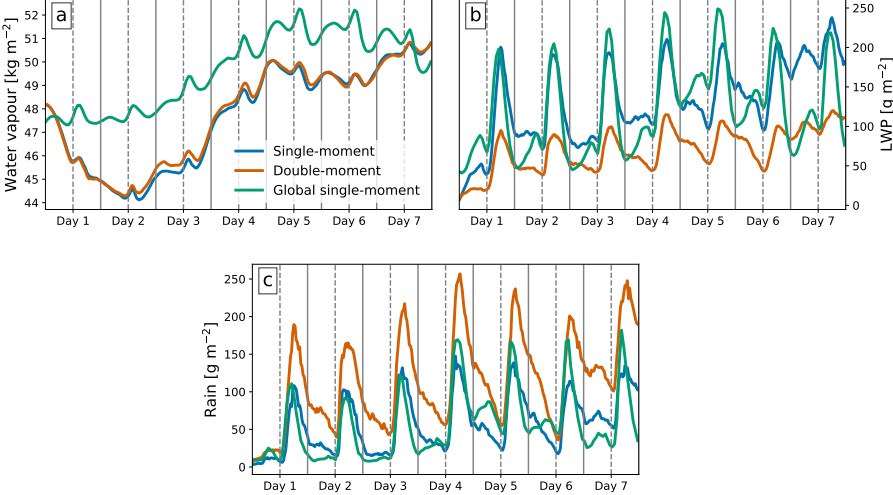

**Figure 8.** Time series of (a) water vapour path, (b) LWP, and (c) rain water path for the regional single-moment (blue) and double-moment (red) simulations, shown alongside the global single-moment run (green). The time series are averaged over the entire Amazon region and experiment duration. Note that the data presented is measured in mass per unit area, where water vapour is shown in $\mathrm{kg\,m^{-2}}$ and liquid/rain water in $\mathrm{g\,m^{-2}}$ for convenience.

with a larger peak-to-peak range compared to the regional runs. Maximum LWP values differ notably; the average difference is $102.7\,\mathrm{g\,m^{-2}}$ between the regional simulations and $16.6\,\mathrm{g\,m^{-2}}$ between the single-moment simulations.

The time-averaged surface precipitation maps for the three simulations are presented alongside IMERG estimates (Fig. 9) to directly compare spatial distribution. The simulations closely match IMERG in spatial distribution, with the northern region remaining relatively dry. Both simulations accurately predict heavier coastal precipitation in the northeast and several organised convection centres, mainly in the south. However, both the single- and double-moment simulations overestimate precipitation intensity compared to IMERG. To enable consistent spatial comparison, the simulated data is regridded to the IMERG grid (approx. $10\,\mathrm{km}$). However, it is important to interpret these discrepancies with caution. Although IMERG provides precipitation data on a fine spatial grid, its effective resolution, i.e., the smallest reliably resolved feature, is coarser than the nominal $0.1^\circ$ grid due to limitations in satellite retrieval capabilities (Huffman et al., 2022). Its accuracy varies by geographic setting and precipitation type; it tends to underestimate heavy rainfall and overestimate moderate events, especially in tropical and mountainous regions, and performs less reliably during winter or in complex terrain (Bulovic et al., 2020; Tan et al., 2017). These biases may partly explain the differences in precipitation intensity.

To further examine the distribution of surface precipitation, Fig. 10 presents the time series of domain-mean surface precipitation (Fig. 10a) alongside its logarithmic scale PDF (Fig. 10b). While the simulated results broadly capture the diurnal cycle observed by IMERG, they tend to overestimate precipitation, often by nearly a factor of two, as on day four. Although precipitation values are generally similar across simulations, subtle differences arise; the global simulation consistently diverges from regional counterparts, particularly at minima and increasingly at maxima from day five onward. Fig. 10b reveals that all simula-





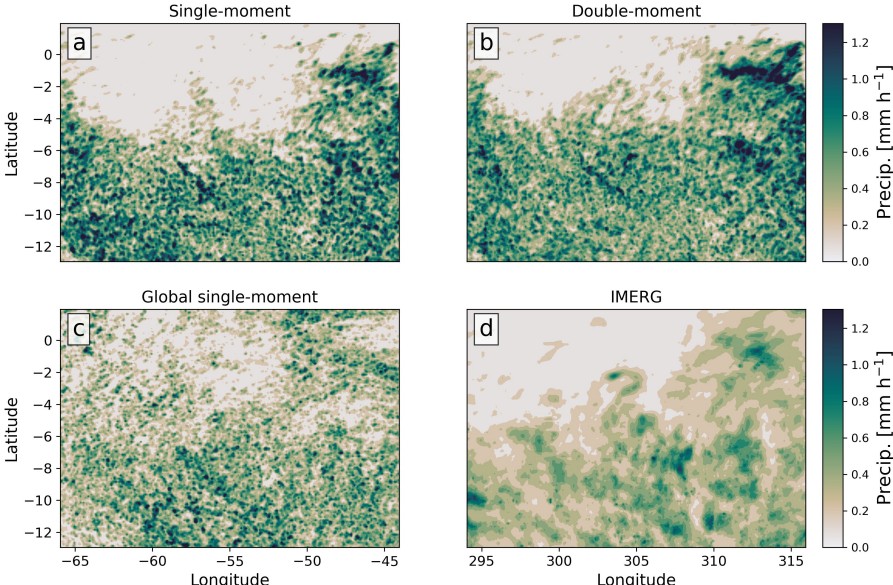

**Figure 9.** Time-averaged maps of surface precipitation rate for the regional (a) single-moment and (b) double-moment simulations, shown alongside (c) the global single-moment run, and (d) IMERG estimates.

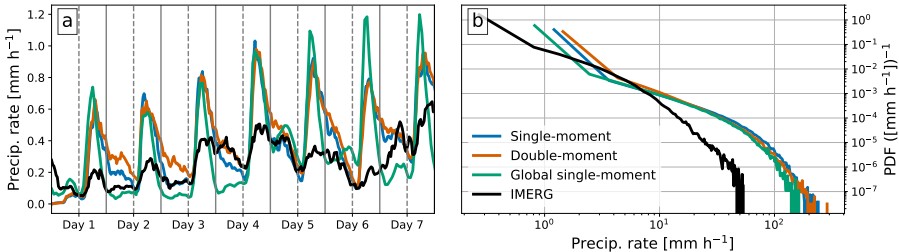

**Figure 10.** (a) Time series and (b) probability density functions of surface precipitation rate for the regional single-moment (blue) and double-moment (red) simulations, shown alongside the global single-moment run (green) and IMERG estimates (black). The time series are averaged over the entire Amazon region and experiment duration.

tions overpredict the frequency of precipitation across most intensities compared to IMERG, especially at higher precipitation rates. The regional simulations exhibit similar distributions, with the double-moment predicting marginally lower probabilities for rates exceeding $150 \, \mathrm{mm \, h^{-1}}$. The global simulation diverges from the regional simulations, particularly at the tail of the distribution, underestimating the probabilities of extreme precipitation. At low rates, IMERG data increase more gradually, indicating that simulations overestimate light precipitation, exposing limitations in capturing the full intensity spectrum. Some differences may stem from IMERG's known constraints in accurately retrieving high-intensity and sub-daily variability, suggesting the observed tail discrepancies reflect both model biases and observational uncertainties.

250



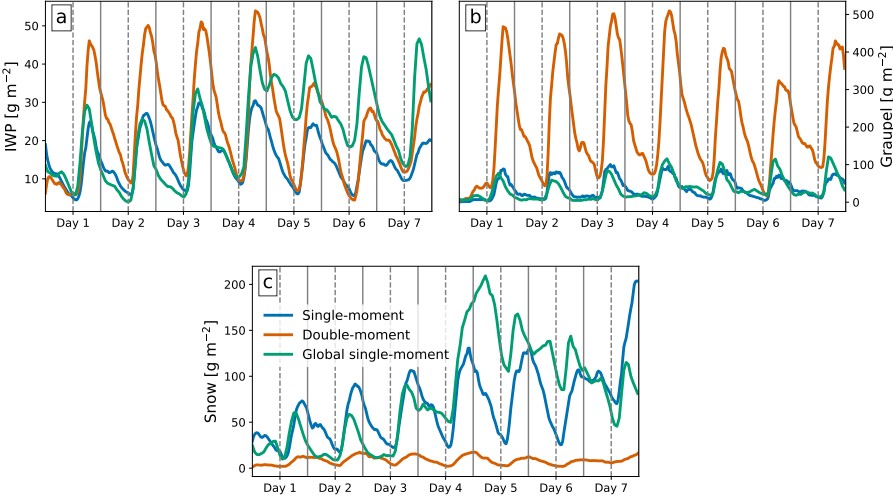

**Figure 11.** Time series of (a) IWP, (b) graupel and (c) snow water paths for the regional single-moment (blue) and double-moment (red) simulations, shown alongside the global single-moment run (green). All variables are expressed in $\mathrm{g\,m^{-2}}$. The time series are averaged over the entire Amazon region and experiment duration.

## 3.3 Ice clouds

Differences in the ice-phase hydrometeors are illustrated by the time series of domain-mean IWP, and graupel and snow water paths from the three simulations (Fig. 11). For IWP (Fig. 11a), the single-moment runs remain closely aligned during the first half of the experiment (days one through four), whereas the double-moment simulation produces nearly twice the peak values. During the second half (days four through seven), IWP in the global single-moment run increases sharply, by about $10\,\mathrm{g\,m^{-2}}$, and exceeds that of the double-moment run, which decreases and approaches the regional single-moment values. The regional single-moment simulation shows a slight decline in IWP after day four, but the change is less pronounced. Graupel behaviour (Fig. 11b) differs substantially across schemes. The single-moment simulations exhibit similar diurnal cycles, although the regional run maintains elevated graupel concentrations slightly longer into the night. The double-moment simulation produces markedly larger graupel amounts, reaching peaks over $400\,\mathrm{g\,m^{-2}}$, about five times higher than the single-moment runs, and maintains high values with little nocturnal relaxation. Snow evolution (Fig. 11c) shows the opposite tendency. The double-moment simulation maintains snow levels nearly constant at low values, around $10\,\mathrm{g\,m^{-2}}$, whereas both single-moment runs produce considerably higher and more variable amounts. During the first half of the experiment, snow in the single-moment simulations ranges between $50$ and $100\,\mathrm{g\,m^{-2}}$; after day four, peak values increase to about $120\,\mathrm{g\,m^{-2}}$ in the regional run and up to $200\,\mathrm{g\,m^{-2}}$ in the global run. The contrasting behaviour of graupel and snow across schemes highlights the differing representations of mixed-phase processes and tuning.

The time-averaged OLR maps from the three simulations are compared with GOES-16 observations (Fig. 12) to assess the simulated cloud coverage. GOES-16 shows suppressed OLR in the southern part of the domain ($200$-$220\,\mathrm{W\,m^{-2}}$), with a





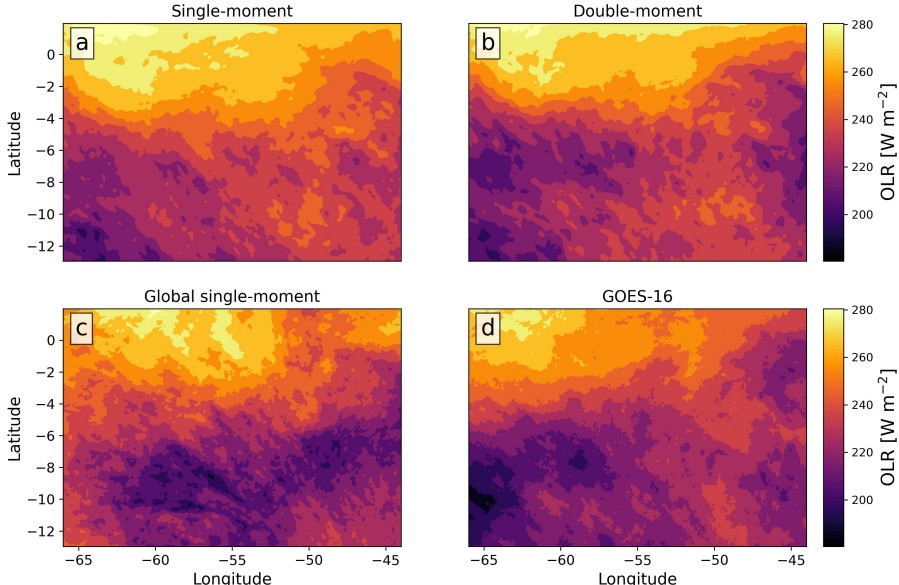

**Figure 12.** Time-averaged maps of OLR for the regional (a) single-moment and (b) double-moment simulations, shown alongside (c) the global single-moment run and (d) GOES-16 observations.

compact low-OLR core extending southeastward and a smooth gradient toward higher OLR over the north. Both regional simulations reproduce the north–south gradient, though with sharper transitions than observed. The double-moment run (Fig. 12b)
captures the location and magnitude of the low OLR core more closely, while the global run (Fig. 12c) extends the suppressed region too broadly. In contrast, the regional single-moment simulation (Fig. 12a) maintains generally higher OLR values and weaker suppression, underestimating deep convection compared to GOES-16. While neither simulation fully reproduces the observations, the double-moment run exhibits better agreement in matching the structure of the suppressed region. In contrast, the global simulation effectively captures the magnitude of this feature, albeit with an exaggerated spatial extent.

The time series of domain-mean OLR in Fig. 13a shows that all simulations capture the diurnal cycle, with peaks and troughs generally aligning with GOES-16. In the first half of the experiment (days one through four), the global simulation maintains higher OLR daily mean values than both regional simulations and GOES-16, then drops from day five onwards. This difference, as expected from the free-running global setup, aligns with the transition seen in IWP (Fig. 11a), suggesting evolving cloud characteristics. Among the simulations, the double-moment model most closely follows GOES-16 retrievals at
minima, typically associated with deep convection. All simulations, however, show higher daytime OLR maxima than GOES-16 (except during days five to seven in the global simulation), indicating shorter-lived or less extensive upper-level clouds, including cirrus and anvil clouds, which can maintain low OLR after convection collapses. From the PDFs shown in Fig. 13b, all simulations predict enhanced probabilities at the high-OLR tail, possibly indicating warm surface biases or reduced high cloud cover. They also show a broader peak around $275\,\mathrm{W\,m^{-2}}$, compared to the narrower GOES-16 peak. A secondary peak
near $220\,\mathrm{W\,m^{-2}}$, potentially representing mid-level convective cloud tops, is captured in both regional simulations, although its



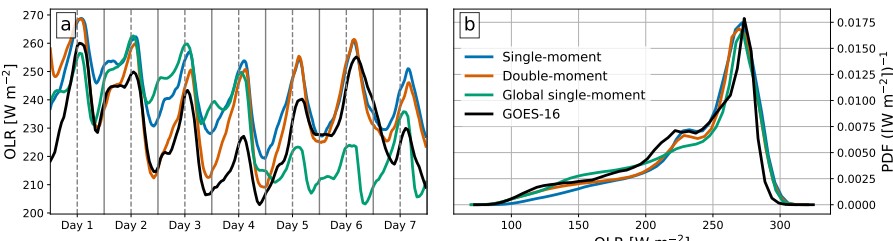

**Figure 13.** (a) Time series and (b) probability density functions of OLR for the regional single-moment (blue) and double-moment (red) simulations, shown alongside the global single-moment run (green) and GOES-16 (black). The time series are averaged over the entire Amazon region and experiment duration.

width is underestimated relative to GOES-16. At lower OLR values (below $150\,\mathrm{W\,m^{-2}}$), indicative of deep convection cloud tops, the single-moment regional simulation underpredicts occurrence; the global simulation aligns closer with the double-moment model, implying similar anvil coverage, consistent with Figs. 4, 11a and 13a.

## 4 Discussion

The simulations clearly demonstrate that the choice of microphysics scheme affects cloud structure, hydrometeor distributions, and precipitation processes, similar to previous studies (see e.g., Khain et al., 2015; White et al., 2017; Igel et al., 2015). However, all simulations produced precipitation patterns that diverge from IMERG estimates (Fig. 9d). While IMERG show convective organisation, all three simulations fail to reproduce this pattern, and predict higher intensities. These discrepancies are consistent with previous findings that km-scale models tend to overestimate rainfall intensity and struggle to reproduce

the degree of convective organisation observed in the Amazon (see e.g., Martins et al., 2015; Pascale et al., 2015). This issue is common in km-scale models but may be improved with smaller grid spacing (see e.g., Moseley et al., 2020; Hohenegger et al., 2020), which is potentially linked to thermodynamic effects of cold pools (see e.g., Tompkins, 2001); however, such resolutions remain computationally challenging for global simulations.

All three simulations also exhibited a more pronounced diurnal cycle than IMERG (Fig. 10a), with clear afternoon and

nighttime peaks. Previous studies report regional variations in Amazon rainfall, including differences in peak timing (see e.g., Angelis et al., 2004). IMERG has been shown to underestimate peak rainfall intensities associated with deep convection (see e.g., Tang et al., 2020; Muñoz de la Torre et al., 2024), which explains its lower maximum values compared to the simulations. These considerations emphasise that comparisons with IMERG should be interpreted cautiously, particularly with respect to the magnitude and timing of extremes.

Section 3 also highlights that, beyond the differences arising from the choice of microphysics scheme, domain configuration also influences simulation sensitivities. Hence, the following discussion considers microphysics sensitivity before addressing domain implications.





## 4.1 Sensitivity to microphysics schemes (single vs. double)

The choice of microphysics scheme influences the simulated cloud processes and hydrometeor distributions. The single-
moment scheme consistently produces higher LWP, fog, and snow values compared to the double-moment scheme, with differ-
ences reaching up to $100\%$, $400\%$, and $1400\%$, respectively (Figs. 7, 8, 11). These differences arise from the way hydrometeor
evolution responds to the underlying microphysical assumptions within each scheme. Overall, these results are consistent with
previous studies, showing that single-moment schemes tend to overestimate low-level cloud condensate and precipitation rates,
particularly in convective environments (e.g., Igel et al., 2015; Naumann et al., 2024; Köcher et al., 2023).

On average, the single-moment simulations produce more warm-phased clouds and thinner, less vertically extended anvil
clouds compared to the double-moment scheme (Figs. 5a and 5b). The reduced anvil cloud cover in the single-moment scheme
may be due to its reliance on fixed-size distributions and riming thresholds, which can suppress effective sedimentation or ice
mass flux aloft. In contrast, the increased flexibility of the double-moment scheme in representing evolving size spectra can
support sustained anvil development. This aligns with previous studies suggesting that changes in size distribution parameters
and number concentrations can influence both precipitation efficiency and the vertical extent of ice clouds (Heymsfield et al.,
2005).

    On the other hand, the single-moment scheme consistently predicts less rain, graupel, and IWP compared to the double-
moment scheme, with observed differences reaching up to $100\%$, $400\%$, and $100\%$, respectively (Figs. 8, 11). The formation of
graupel, a key contributor to precipitation efficiency, relies on the accretion of cloud water, which is treated with greater process-
level flexibility in the double-moment scheme. Additionally, differences in warm-rain processes, such as autoconversion and
collection efficiencies, likely contribute to the observed discrepancies in precipitation (Fig. 9), with studies showing that double-
moment schemes more accurately represent the sensitivity of rain formation to variations in droplet size and concentration,
which can modulate rainfall efficiency across different convective regimes (Zhang et al., 2018). Despite this, the two schemes
yield similar levels of domain-averaged precipitation, OLR, and water vapour (Figs. 10, 13, and 8). This indicates that although
microphysical processes influence the distribution and characteristics of hydrometeors, larger-scale dynamics, which primarily
drive these outputs, are less sensitive to the choice of microphysics scheme. This conclusion is consistent with Dagan et al.
(2019), which demonstrated that at large spatial scales (where precipitation approximates evaporation), changes in precipitation
are constrained by water and energy budgets. Conversely, at smaller scales, the divergence of water vapour (i.e atmospheric
dynamics) can predominantly govern variations in precipitation.

Interestingly, there are cases where the single-moment scheme simulation has demonstrated closer approximations to ob-
served storm characteristics, such as in simulations of Super Typhoon Sarika (2016), where single-moment schemes produced
a stronger storm than double-moment schemes (Li et al., 2020). These findings highlight that the relative performance of mi-
crophysics schemes may depend on the specific meteorological context and the processes dominating the cloud system under
consideration.

However, it is important to address additional limitations associated with the choice of microphysics schemes. Double-
moment schemes require greater computational load due to the larger set of prognostic variables they introduce (typically



twice as many). Although this enables a more detailed microphysical representation, it does not necessarily improve the accuracy of predicted integrated quantities, such as precipitation or radiative fluxes (Köcher et al., 2023). Beyond computational cost, both schemes rely on tuning parameters that influence their simulated behaviour and may contribute to the observed divergence. Although tuning can improve agreement with observations, it reduces generalisability across regimes and complicates direct comparisons between schemes. While these aspects are beyond the scope of this study, they should be considered when interpreting the results.

## 4.2 Implications of domain configuration (global vs. regional)

While microphysics schemes primarily drive differences in cloud structure and hydrometeor distributions, the domain configuration modulates these sensitivity manifestations. The global single-moment run produces higher values of cloud ice, fog, precipitation, and water vapour than the regional single-moment run, with differences reaching up to $20\%$, $150\%$, $30\%$, and $10\%$ respectively (Figs. 5, 7, 8, 10). These contrasts reflect the influence of large-scale circulation and energy transport, which are shaped by the domain size and the treatment of boundary conditions (Dagan et al., 2022b, a). Whereas the regional setup constrains circulation and thermodynamic profiles through sponge layers and prescribed boundary conditions, the global configuration allows the development of internally consistent synoptic-scale convection, enhancing moisture transport and convective organisation.

Temporal evolution highlights how the differences accumulate over time, with similar initial IWP values between the single-moment simulations and nearly doubled peak values in the global run during the second half of the experiment (Fig. 11a). This deviation, independent of the initial state (Fig. 2), points to the growing impact of boundary effects on cloud properties and moisture structure as simulations progress, consistent with studies showing that convective triggering and intensity, boundary-zone divergence and large-scale advection are sensitive to boundary conditions (Rybka et al., 2021).

Fog formation provides a clear example of this interaction between microphysics and domain-scale dynamics. Although fog is mainly influenced by the microphysics scheme, the higher near-surface water vapour in the global simulation (Fig. 5f) promotes a more persistent fog (Fig. 7), likely due to weaker boundary-layer ventilation and reduced horizontal advection. This persistence is reinforced by a feedback mechanism in which fog reduces surface heating, slows vapour removal, and sustains itself under inversion capping until dissipated by solar forcing. Such processes illustrate how domain-scale dynamics can amplify or prolong microphysics-induced variability (Anber et al., 2015; Tardif, 2017). The simultaneous increase in IWP and decrease in OLR during the latter part of the global simulation further suggests the onset of deeper convection, which eventually reduces fog through enhanced vapour removal.

It is important to consider that the longer timestep used in the global simulation ($40\,\mathrm{s}$ vs. $10\,\mathrm{s}$ regionally) likely influences process rates and biases, such as fog and warm-cloud formation, thereby contributing to further divergence between configurations (Schmidt et al., 2023).

Taken together, these results demonstrate that, while the microphysics primarily controls cloud structure and hydrometeor characteristics, domain size shapes the dynamical environment in which these sensitivities are expressed. Global simulations enable the growth of large-scale circulations and moisture transport, which can amplify or reshape microphysics-driven differ-



ences, thus reinforcing the broader conclusion that integrated outputs, such as precipitation and OLR, are ultimately governed by this large-scale dynamical framework.

## 5 Summary and conclusions

In this study, we investigate how simulated cloud and precipitation properties respond to different microphysics schemes
and how these responses manifest in regional and global configurations in ICON. Three convection-permitting runs over the Amazon basin are analysed: a global simulation with a single-moment cloud microphysics scheme and two regional simulations with single- and double-moment schemes, with other parameters held constant.

Results show that microphysics schemes exert the strongest influence on cloud hydrometeors. Compared to the regional single-moment run, the double-moment scheme produces up to five times more graupel and about twice as much rain and IWP,
but up to twice less LWP, five times less fog and an order of magnitude less snow (Figs. 7, 8, 11). These contrasts likely reflect the explicit treatment of number concentrations and size distributions in the double-moment scheme, which affects processes such as autoconversion, sedimentation, and riming. At the same time, scheme-specific tuning parameters may contribute, meaning that part of the discrepancy arises from parameter choices rather than scheme structure alone. Despite these large microphysical differences, domain-averaged precipitation, water vapour, and OLR are similar across schemes (Figs. 8, 10, 13),
indicating that large-scale dynamics and budget constraints dominate integrated atmospheric metrics.

Domain configuration further controls how these sensitivities manifest. The global single-moment run contains up to $150\%$ more fog, nearly twice the IWP, and roughly $10\%$ more water vapour than its regional counterpart (Figs. 7, 8, 11). These variations reflect the influence of domain size and boundary treatments on circulation, moisture transport, and convective organisation. After day four, the global run shows a sharp decline in fog, a doubling of IWP, and a reduction in OLR (Figs. 11,
13), changes absent in the regional runs. This behaviour highlights the ability of global simulations to develop internally consistent circulation patterns, whereas regional setups remain constrained by their boundaries. Additional divergence may also stem from the longer timestep used in the global run, which could affect fog and warm-cloud formation.

However, several limitations should be noted. Only one global simulation is performed, preventing direct comparisons between single- and double-moment schemes, as global km-scale double-moment runs remain computationally demanding.
The observational data also carry uncertainty, as IMERG tends to underestimate extreme rainfall and independent datasets are scarce. This highlights the ongoing need for detailed and consistent observations to better constrain cloud and precipitation processes. Both microphysics schemes involve numerous tunable parameters that add further degrees of freedom. Perturbed-parameter ensembles (PPEs) are therefore needed to disentangle structural and parametric uncertainty and to quantify the sensitivity of simulated cloud processes to parameter choices. Global double-moment simulations, combined with PPEs, would
help clarify how domain size, circulation, and microphysics interact to influence hydrometeor development.

In summary, km-scale regional simulations capture many aspects of cloud microphysics but may miss interactions such as moisture recycling, remote convection, and large-scale advection. Global simulations couple local and large-scale processes, but require higher computational demand and longer spin-up times. With global convection-permitting modelling now op-



erational, progress depends on combining improved observational constraints with ensemble experimentation to refine the
representation of clouds and reduce remaining uncertainties.

*Code and data availability.* The simulations were done using the open-source ICON model described by Hohenegger et al. (2023). Access to the ICON source code for scientific use is available from https://code.mpimet.mpg.de/projects/iconpublic (last access: 6 October 2025). The code, including the model configurations, is provided in Sela et al. (2025). ERA5 reanalysis data were obtained from the Copernicus Climate Data Store (https://doi.org/10.24381/cds.adbb2d47, Hersbach et al. (2023)). The Integrated Multi-satellitE Retrievals for GPM (IMERG) v06 precipitation data (NASA, 2020) were obtained from the NASA Goddard Earth Sciences Data and Information Services Center (GES DISC) at https://gpm1.gesdisc.eosdis.nasa.gov/data/GPM_L3/GPM_3IMERGHH.06 (last access: 6 October 2025). Outgoing longwave radiation (OLR) was derived from Level 2 data of the Advanced Baseline Imager (ABI) aboard GOES-16 (Schmit and Gunshor, 2020), obtained through NOAA's Comprehensive Large Array-data Stewardship System (CLASS; https://www.class.noaa.gov, last access: 6 October 2025). Product documentation for the ABI OLR algorithm is available at https://www.star.nesdis.noaa.gov/GOES (last access: 6 October 2025) and in Lee et al. (2010). Model, observation and reanalysis data used in this paper are available from Sela (2025) (https://zenodo.org/records/17592760).

*Author contributions.* MS and PW obtained the simulations under the supervision of PS. MS performed the analysis and prepared the manuscript with contributions from all of the co-authors.

*Competing interests.* The authors declare that they have no competing interests.

*Acknowledgements.* The simulations were performed and analysed on the Levante cluster of the DKRZ with resources granted under project 1368 (https://www.dkrz.de/en/systems/hpc/hlre-4-levante, last access: 6 October 2025). Maor Sela acknowledges funding from the NERC Doctoral Training Partnership in Environmental Research Grant NE/S007474/1. Philipp Weiss and Philip Stier acknowledge funding from the European Union's Horizon 2020 project nextGEMS under grant agreement number 101003470 and Philip Stier from the European Union's Horizon Europe project CleanCloud with grant agreement 101137639 and its UKRI underwrite.



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
