# Peer review of "Sensitivity of cloud structure and precipitation to cloud microphysics schemes in ICON and implications for global km-scale simulations"

_EGUsphere, 2025_

## Referee Comment (RC1)

Review of "Sensitivity of cloud structure and precipitation to cloud microphysics schemes in ICON and implications for global km-scale simulations"
Author(s): Maor Sela, Philipp Weiss, and Philip Stier
Submitted to GMD
MS No.: egusphere-2025-5803

**Recommendation:**

Major revisions

**General comments:**

This article examines the sensitivities of two cloud microphysics schemes—a single-moment bulk scheme and a double-moment bulk scheme—implemented in the regional configuration of ICON, as well as the sensitivities associated with two domain configurations: a regional model and a global model. The target region selected by the authors is the Amazon over land, and the simulations are conducted for a one-week period.

The motivations for examining these sensitivities are well justified, and similar sensitivity studies can be found throughout the existing literature. However, the conclusions drawn in this study may depend on the specific implementations of the cloud microphysics schemes, as well as on the choice of domain and simulation period. Consequently, it is unclear to what extent these conclusions can be generalized to other cases.

Despite these limitations, the authors clearly describe the simulation results, and the article can serve as a useful reference for understanding such dependencies. Therefore, the reviewer recommends that this article be considered for publication, provided that the authors address the comments raised below adequately.

In addition to the cloud microphysics profiles, the authors analyze the characteristics of fog. However, in this article, the simulated fog is not compared with observations, making it unclear which results are more realistic. The reviewer therefore requests that the authors include an evaluation of the simulated fog against observational data. Fog formation is generally controlled by thermodynamic conditions rather than by the detailed formulation of cloud microphysics, with surface temperature and humidity near the surface being the most influential factors. The dynamic and thermodynamic representations in the model should be examined in detail.

The differences between the regional and global model results appear to arise primarily from how large-scale (synoptic-scale) circulations are represented in the global model. In the regional model, these circulations are constrained by the lateral boundary conditions. Therefore, it is important for the authors to examine how the large-scale fields are simulated in the global model and how they may deviate from reality. To this end, the reviewer requests that the authors analyze the large-scale evolution over a domain broader than the target Amazon region. Equatorial waves might represent the modulation of the large-scale fields. A Hovmöller diagram along a representative latitude would likely be sufficient for this purpose.

**Specific comments:**

Introduction:

Although the references cited in the introduction are extensive, the reviewer suggests the following additional references, which would help enrich the evidence presented in the article.

Performance of single vs double moment bulk microphysics schemes in different configuration of the models (L35-42):

- Satoh, M., Matsugishi, S., Roh, W., Ikuta, Y., Kuba, N., Seiki, T., Hashino, T., Okamoto, H. (2022) Evaluation of cloud and precipitation processes in regional and global models with ULTIMATE (ULTra-sIte for Measuring Atmosphere of Tokyo metropolitan Environment): A case study using the dual-polarization Doppler weather radars. Progress in Earth and Planetary Science, 9, 41.https://doi.org/10.1186/s40645-022-00511-5

For uncertainties of cloud microphysics profiles across global storm-resolving models (L68-78):

- Roh, W., Satoh, M., Hohenegger, C. (2021) Intercomparison of cloud properties in DYAMOND simulations over the Atlantic Ocean. J. Meteorol. Soc. Japan, 99, https://doi.org/10.2151/jmsj.2021-070

L59: "Regional CRMs" are the central part of the operational short-range forecast. This aspect of the operational model activities should be described here.

L75: "a steady state" should be "a statistically steady state" or "a quasi-steady state".

L139: It is unclear why different time steps are used for the regional and global models, and in particular why the time step for the regional model is shorter than that of the global model. In general, one would expect the time step of a global model to be shorter than that of a regional model, because the global model is subject to a wider range of topographic and meteorological conditions.

Figures 2 and 3: The authors are requested to add the corresponding figures for the regional model using the two different cloud microphysics schemes. In particular, the representation of surface temperature warrants further examination.

Figure 2 presents a single instantaneous snapshot and therefore does not adequately illustrate the overall agreement or discrepancies between the global model and observations. A more focused analysis over a broader temporal and spatial range is needed to demonstrate how the global model simulates the evolution of large-scale fields. One possible approach would be to include a Hovmöller diagram of convective systems, as requested in the general comments above.

L173-174: As for the limitation of IMERG, it is generally less sensitive to weaker precipitation, including drizzle. The authors must mention this limitation and should be cautious when evaluating the PDF of weaker precipitation (Fig. 10b).

L186, "All simulations predict more clouds": It is unclear what "clouds" mean here.

3.1 Fog: It is unclear why the authors chose fog as a primary metric for evaluating the model results. Fog formation is more strongly constrained by thermodynamic conditions than by cloud microphysical processes, with surface temperature and humidity near the surface being key controlling factors. Moreover, the manuscript does not demonstrate how the simulated fog is compared with observations.

The reviewer therefore requests that the authors first provide observational evidence of fog characteristics during the simulation period. Subsequently, the simulations of surface temperature and low-level moisture—both critical factors for fog formation—should be evaluated in detail.

L206-207, "···fog is more sensitive to microphysics choice": Instead of this, fog is more sensitive to surface temperature and humidity near the surface. The choice of microphysics is of secondary importance.

L243, Figure 8c vs Figure 10a: Please explain the relation between Fig. 8c "rain" and Fig. 10a "precipitation". Does Figure 8c show the column integral of the mixing ratio of rain particles? Why does it evolve differently with precipitation? Figure 8 shows that the double moment scheme is an outlier, while Figure 10 shows the global model outstands with the regional model.

L252, "indicating that simulations overestimate light precipitation": This may also be because IMERGI is less sensitive to weak precipitation.

L300: Please refer to the following literature:
· Rehbein, A., Ambrizzi, T. (2023) Mesoscale convective systems over the Amazon basin in a changing climate under global warming. Clim. Dyn. https://doi.org/10.1007/s00382-022-06657-8

L316: Please refer to Fig. 5 in addition to Figs. 7, 8, 11.

L336-339, "This conclusion is consistent with Dagan et al. (2019), which demonstrated that at large spatial scales (where precipitation approximates evaporation), changes in precipitation are constrained by water and energy budgets."; The present results cover only a short time range (within one week), for which precipitation does not approximate evaporation. Therefore, it is not appropriate to suggest consistency based on these budgets.

L367-368, "Although fog is mainly influenced by the microphysics scheme": Fog is controlled by thermodynamic processes near the surface. The microphysics scheme is of secondary importance.

L373-374, "The simultaneous increase in IWP and decrease in OLR during the latter part of the global simulation further suggests the onset of deeper convection": Throughout this article, the onset of deeper convection is not clearly demonstrated for any cases considered, making this statement inappropriate.